# Efficient Certificate-Less Aggregate Signature Scheme with Conditional Privacy-Preservation for Vehicular Ad Hoc Networks Enhanced Smart Grid System

**DOI:** 10.3390/s21092900

**Published:** 2021-04-21

**Authors:** Thokozani Felix Vallent, Damien Hanyurwimfura, Chomora Mikeka

**Affiliations:** 1African Center of Exellence in Internet of Things (ACEIoT), College of Science and Technology, University of Rwanda, KN Street Nyarugenge, Kigali P.O. Box 3900, Rwanda; tfvallent@gmail.com (T.F.V.); dhanyurwimfura@ur.ac.rw (D.H.); 2Department of Physics, Chancellor College, University of Malawi, Zomba P.O. Box 280, Malawi

**Keywords:** smart grid, vehicular ad hoc network, privacy-preserving, certificate-less, signature, aggregation

## Abstract

Vehicular Ad hoc networks (VANETs) as spontaneous wireless communication technology of vehicles has a wide range of applications like road safety, navigation and other electric car technologies, however its practicability is greatly hampered by cyber-attacks. Due to message broadcasting in an open environment during communication, VANETs are inherently vulnerable to security and privacy attacks. However to address the cyber-security issues with optimal computation overhead is a matter of current security research challenge. So this paper designs a secure and efficient certificate-less aggregate scheme (ECLAS) for VANETs applicable in a smart grid scenario. The proposed scheme is based on elliptic curve cryptography to provide conditional privacy-preservation by incorporating usage of time validated pseudo-identification for communicating vehicles besides sorting out the KGC (Key Generation Center) escrow problem. The proposed scheme is comparatively more efficient to relevant related research work because it precludes expensive computation operations likes bilinear pairings as shown by the performance evaluation. Similarly, communication cost is within the ideal range to most related works while considering the security requirements of VANETs system applicable in a smart grid environment.

## 1. Introduction

Major advancement in wireless sensor networks (WSN), Internet of Things (IoT) and the advent of the big data paradigm has seen the birth of various network based advancements in cross-cutting technologies, such as VANETs, which support wireless communication within vehicles and road sign units (RSUs) for numerous applications like traffic safety, location based-services, electric vehicles (EVs) and electricity exchange services among others [1,2,3,4,5,6]. The smart grid is one such technology motivated by the development of WSN and IoT in its functionality. EV technology will result in the elevation of power consumption, unsustainable by means of a traditional electricity grid [7]. An obvious solution to sorting out EVs electricity demands is by formulating VANETs-enhanced smart grid, with a coordinated charging system that is responsive to efficient cost and electricity utilization by using communication technologies [8,9]. Thus, it is recommended that algorithms for security, authentication, information processing and data aggregation be of high-precision and efficiency to allow low communication latency for real-time pricing and optimal electricity dispatch decisions in a VANETs enhanced smart grid system [10,11]. The concept of VANETs is an advancement of mobile ad hoc networks (MANETs) where there is real-time communication between EVs and RSUs for electricity charging/discharging [7,12,13]. Typically, the topology of VANETs includes trusted authorities (TAs), RSUs and onboard Units (OBUs) mounted on vehicles [14,15,16]. The OBUs constantly cast the traffic related messages about vehicles facilitating various smart applications and technologies such as current vehicle location, time, speed, direction and traffic condition in every 100–300 ms [13,17,18]. As is the case with many communication network based technologies, VANETs is not an exception to face various cyber-security challenges in terms of data security and user privacy [19,20,21,22]. With secure and privacy protection addressed, the applications of VANETs in traffic management and control, traffic accident avoidance features, traffic vigilance, gas emission, EV charging and fuel consumption will be fully implementable [23]. So if the VANETs network system is not protected, adversaries may launch all sorts of attacks like data modification, impersonation, replay, denial of service attacks among others. For instance, there are attacks launched by rogue vehicles broadcasting fake instructions to cause traffic accidents and general confusion. Thus, in terms of message senders’ legitimacy there should be security features when sending messages to check authentication and integrity [23,24]. To this effect many authentication schemes have been proposed using traditional public key cryptography (PKC) to secure a VANETs system [25,26]. In terms of privacy concerns, anonymity must be provided in the design of securing the message from an eavesdropping adversary. In this way the real identity of communicating party will not be known nor communication transactions be analyzed and linked to a particular VANETs participant. However, due to abuse of the anonymity feature, the pseudonym given to participating entities should be traceable and revocable, so that the TA can reveal the real identity of malicious vehicle under certain conditions [27]. Since OBUs have limited computation and storage capabilities, the use of less computation intensive cryptographic techniques is promoted so as to handle large message flow in the system and improve smooth communication. Certificate-less aggregate signature (CLAS) is one efficient technique that improves message authentication by utilizing batch calculation of which saves bandwidth. In CLAS *n* signatures on *n* distinct messages from *n* distinct users, are aggregated into a single short signature that can be verified at once as combined [28]. This approach is very helpful in VANETs where RSUs collects and aggregate a large number of signatures from individual participants signatures into one signature that is broadcasted to vehicles in the system to achieve a particular VANETs enhanced smart grid application, and this greatly enhances efficiency in verification and communication overhead [13,29]. Achieving efficiency by design is much encouraged to cope up with the computation capabilities of RSUs and OBUs by constructing the algorithms with lighter computation operations. To this effect, employing elliptic curve cryptography (ECC) based cryptosystems improves computation efficiency by a great margin and thereby a recommendable approach. Thus, we propose an efficient certificate-less aggregate scheme with conditional privacy-preservation by using the ECC approach. The proposed scheme satisfies security and privacy requirements for VANETs with optimal efficiency and rigorous security proof is provided. There are different modes of communications in VANETs such as vehicle-to-vehicle (V2V), vehicle-to-grid (V2G) and vehicle-to-infrastructure (V2I) and vehicle-to-everything (V2E), which use the short medium range communication protocol called dedicated short range communication (DSRC) to facilitate various vehicular network applications [30]. These computer sophisticated vehicles are being adopted for various smart services in intelligent transportation systems (ITS). The following security requirements are important for any WSN based system such as VANETs:Non-repudiation: Any electric vehicle transaction has economic value and this can motivate fraudulent acts by the entities selling or buying electricity. Therefore, this measure of non-repudiation ensures that any electricity transaction can be accounted for, to the involved parties and any modification cannot be denied by the party.Message integrity and authentication: In a similar manner, any network transaction once completed cannot be modified by any malicious entity and once there is an attempt to tamper with the transaction, then it should be detectable by any legal entity of the system.Privacy: The actual identity of a consumer nor the information of a transaction in the network should not be known by any malicious party eavesdropping on the communications involving a particular targeted entity.Unlinkability: By observing the transactions in the VANETs network the entity’s activities should still not be analysed and be associated with a particular RSU or vehicle. Thus to say messages plying on the network for any participant should still look random to an attacker and nothing associated with the participant should be determined.Traceability: However, for the undesirable conduct of an entity in the network such acts should be traced and be accounted for, against the individual. On the other hand the vehicle should be hidden or inaccessible from other unauthorized entities.Resistance to Attacks: Due to communication over a public channel, V2G security scheme must withstand various general attacks such as an impersonation attack, replay attack, modification attack, man-in-the-middle-attack and stolen verifier table attack in VANETs.

Therefore, we propose a novel anonymous certificate-less aggregate signature scheme for VANETs with conditional privacy-preservation in a smart grid system, that addresses common weaknesses of most existing certificate-less aggregate signature schemes. The main contribution of the paper can be summarized as follows:The proposed scheme achieves user anonymity with conditional privacy, such that each domain stores a Certificate Revocation List (CRL) in all road sign units located in that particular domain.The proposed scheme achieves optimal efficiency for certificate-less aggregate signature while precluding complex cryptographic operations like bilinear pairings and map-to-point hash operations.The proposed scheme withstand escrow property powers of the KGC but use of partial private key and user generated full private key for signature signing.

The rest of the paper is organized according to the outline given as follows—Section 2 reviews most relevant related works of CLAS schemes for VANETs. Section 3 provides the mathematical building blocks for the proposed scheme. Section 4 gives the detailed steps of the proposed work. Section 5, presents an indepth analysis of the scheme in terms of security, privacy and performance assessment. Finally, in Section 6 we give concluding remarks about the proposed scheme.

## 2. Related Works and Limitations

In VANETs, the source authentication and message integrity of traffic-related information form a very important security requirement in the system. Satisfaction of these security requirements ensure the trust and proper functionality of all versatile technologies that comes with a VANETs system by simply securing moving vehicles, RSUs, Application Servers, and roadside sensors. To this effect many research works have been done to provide the needed security for such an advent technology of smart city [24].

The key management problem posed by the certificate based PKI cryptosystem paved the way to the pioneering work of a certificate-less public key signature (CL-PKS) scheme by Al-Riyami and Paterson [31]. This idea caught much research interest in the aspect of improving the security and performance. In [32], Yum and Lee presented a general procedure to construct a CL-PKS scheme from any ID-based signature scheme. The first CL-PKS scheme was bilinear pairing based proposed by Li et al. in [33]. Whereas in [34], Au et al. presented a new security model for CL-PKS schemes which considers inside attack scenario. The first bilinear pairing free CL-PKS scheme was first proposed by He et al. in [35], which was found to be vulnerable to other attacks in [36]. In [37] a scheme ideal for IoT deployment was proposed; however, it was found to bear some flaws concerning inside attack performance by KGC in [38]. In order to provide the needed security property of anonymous authentication in [39,40] the idea of pseudonym-based authentication was employed. Despite providing privacy preservation, the limitation of overburdened TA in storing these pseudonyms for each vehicle was encountered as has shown out as the shortfall for their approach. In [41], having foreseen the problem of overburdened TA and sought to provide a solution they designed a scheme by using anonymous certificates but this was done at the expense of interactions between the infrastructures. In [42] et al., privacy protection for VANETs communications was achieved based on the technique of ID-based ring signature, but they failed to provide conditional privacy, since there was no any tracking mechanism in their algorithm [43]. Many more researchers demonstrated the need to formulate robust schemes in terms of security and privacy protection. To this cause, Bayal et al. [44] proposed an anonymous authentication scheme, however it is deemed computationally intensive in [45]. In [46], Cui et al. proposed a scheme that utilizes the methods of a cuckoo filter and binary search to facilitate batch verification for vehicular communication of V2V and V2I. He at al. [17] designed an ECC based certificate-less based signature scheme for VANETs system with batch verification feature. However, Mahmood et al. [31] states that their scheme still vulnerable to side-channel attack since some of sensitive information like TA’s master private key is stored in a tamper-proof devices (TPD). A scheme in [47] uses pseudonyms instead of real identities in trying to secure VANETs communications. The scheme in [47] achieves efficiency and provides batch verification but falls short in terms of providing all security requirements like unlinkability.

## 3. Preliminaries

Now we will formalize the background knowledge of the building blocks for the proposed scheme. The notations used in the designed algorithm are given and described in Table 1. ECC is a public key cryptosystem based on elliptic curve theory and has an advantage for being a structure for faster and more efficient cryptosystems with robust security. ECC cryptosystems have low computational requirement hence its viable for securing resource constrained network systems that require seamless and real-time operations like the IoT and SG systems [48].

*Elliptic curve*: Given a prime number *q*, equation $y^{3}=x^{2}+ax+bmodp$ defines an elliptic curve over a prime field $E(F_{p})$, where $p>3,a,b\in F_{q}$ and satisfies $▵=4a^{3}+27b^{2}\neq 0modp$. The points on $F_{p}$ together with the point at infinity O form an additive cyclic group *G*. Let *P* be the generator point of order *n*, the scalar multiple operation is defined as, nP=P+P+⋯+P, *n* times addition, where $n\in Z_{q}^{*}$, is a positive integer. So, there are a number of intractable problems in an elliptic curve group *G* of order *n*, suitable for cryptographic purposes as there is no polynomial algorithm to solve them efficiently by brute-force within probabilistic polynomial time.

*Elliptic Discrete Logarithm (ECDL) Problem*: Given an element Q∈G, the ECDL problem is to extract an element $x\in Z_{q}^{*}$, such that Q=xP.

*Elliptic Curve Computational Diffie-Hellman (ECCDH) Problem*: Given two elements xP,yP∈G, with unknown elements $x,y\in Z_{q}^{*}$, the ECCDH problem is to compute Q=xyP.

*Elliptic Curve Decisional Diffie-Hellman (ECDDH) Problem*: No any probabilistic polynomial time algorithm can distinguish between the tuples $\left(P_{1},xP_{1},yP_{1},T\right)$ and $\left(P_{1},xP_{1},yP_{1},xyP_{1}\right)$ where $P_{1},T\in G$, with unknown elements $x,y\in Z_{q}^{*}$.

### 3.1. System Model

In terms of the communication process, the VANETs’ architecture is categorized into two layers, namely the physical layer and the application layer, in which case the physical layer is comprised of the vehicles, the RSUs situated on designated points of the road. Vehicles on the roads are embodied with OBUs as a communication enabling device to connect with other vehicles, RSUs or other advanced smart city facilities. [49]. The OBU is equipped with a TPD device to secure stored sensitive information like secret key and the global positioning system (GPS). As such the vehicle is securely able to carry out advanced VANETs communications in smart cities including V2X, V2V and V2I, which are enabled by a dedicated short range communication (DSRC) protocol specifically identified as IEEE 802.11p. On the other hand, the application layers are comprised of the key generation center (KGC) and the tracing authority (TRA) application server, which are the major components undertaking the TA roles in a conditional privacy preserving VANETs based system. The design and the interplay of these main entities in the system is illustrated in Figure 1, where close range networks are facilitated by wireless communication technology such as IEEE802.11p, mid-way network communication is aided by long range communication technology coupled with high bandwidth such as WiMax. Whereas, the backbone network system is empowered by wired communication which is mostly assumed to be secure as it controlled by the public utility company. It is the wireless communication that is supposed to be secured by security algorithm that ensures authentication and integrity on all communications amongst the concerned entities. The TRA is the responsible authority for RSUs and issuing pseudo-identities to vehicles, and can do real identity revocation whenever necessary. In a like manner, the KGC is responsible for public and partial private keys’ generation for both RSUs and vehicles. So in VANETs schemes, it is usually assumed that the KGC and TRA are trusted parties and hence assumed honest but curious [50]. Both KGC and TRA have sufficient computation power but the OBUs and RSUs are the one with limited computation and storage capabilities hierarchically with RSUs as most powerful one [23,29,51]. However, OBUs and RSUs are not trusted entities and therefore any communication initiative originating from them must be authenticated. Thus, this inspires the devising of security protocols for VANETs with suitable computation requirements for OBUs and RSUs.

### 3.2. Security Model for CLAS Scheme

As proposed first in [31], in CLAS we assume two types of adversaries termed *Type 1 Adversary*, $A_{1}$, and *Type 2 Adversary*, $A_{2}$. Here, $A_{1}$ acts as a dishonest user and $A_{2}$ acts as a malicious KGC on the other hand. ***Type 1 Adversary***: $A_{1}$ adversary does not control the master key but is allowed to replace public keys at will, with any desirable value of its choice. ***Type 2 Adversary***: $A_{2}$ adversary has access and controls the master key but cannot replace the public keys of users.

The classical security model proposed in Zhang et al. [52] presents a security adversarial model for certificate-less key agreement schemes. The model is defined as a game between a challenger, *C*, and an adversary defined by a probabilistic polynomial-time Turing machine,$A\in \{A_{1},A_{2}\}$. Thus, *A* has full control of the communication channel of all parties and parties only respond to queries from *A* and cannot communicate directly with each other. As a controller of the communication channel, *A* has powers to actively carry out the following actions, such as relaying, modifying, delaying, interleaving, deleting all the messages flowing in the system.

## 4. The Proposed Certificate-Less Aggregate Signature Scheme

In this section, we will explain the scheme design for VANETs integrated smart grid system titled Efficient Certificate-less Aggregate Signature Scheme with Conditional Privacy-Preservation for Vehicular Ad Hoc Networks Enhanced Smart Grid System. For easy referencing the scheme will be termed ECLAS. The proposed scheme consists of eight algorithms which are: Set-up, Pseudo-Identity Generation, Partial-Private Key Extraction, Vehicle-Key Generation, Sign, Individual Verify, Aggregate and Aggregate verify, which are explained in details as follows.

1.
*Set-up*
In this section, the TA, comprising of two mutually exclusive principle parts, which are the TRA and the KGC, will initialize the system by generating the system parameters. The TA takes as input the security parameter $1^{k}$ the algorithm outputs two large prime numbers, *p*, *q* and a non-singular elliptic curve defined by $y^{2}=x^{3}+ax+b\left(mod\;p\right)$, where $a,b\in F_{p}$.The KGC sets a point *P* from *E* and with this point generates a group *G* of order *q*. Then KGC randomly selects a number $\alpha \in Z_{q}^{*}$ and sets it as its master secret with its corresponding public key computed as $P_{pub}=\alpha P$.Similarly, the TRA selects a points *P* on *E* and with it generates a group *G* of order *q*. Further, TRA chooses a random number $\beta \in Z_{q}^{*}$ and computes its public key $T_{pub}=\beta P$ while setting β as its master secret key used for traceability which is known to TRA only.All these principle entities (TA, KGC and TRA), choose three hash functions, $H_{1}:G\to Z_{q}^{*}$, $H_{2}:{\left\{0,1\right\}}^{*}\to Z_{q}^{*}$ and $H_{3}:{\left\{0,1\right\}}^{*}\to Z_{q}^{*}$Then the system public parameters $params=\{P,p,q,E,G,H_{1},H_{2},H_{3},P_{pub},T_{pub}\}$ are published.These params are then preloaded in the tamper-proof communicating devices and RSU of the system.2.
*Pseudo-Identity-Generation\Partial-Private-Key-Extraction*
In this phase, the TRA’s responsibility is to generate pseudo-identities for the vehicles while the KGC’s responsibility is to create corresponding partial private keys to the pseudo-identities. Thus, finally all vehicles under a TA are registered and preloaded with their pseudo-identities and partial private keys. By use of pseudo-identities that are closed linked to the real identities, the proposed scheme can achieve conditional privacy-preservation when it is necessary to revoke the real identity of an entity the TRA can ably do so. The process of pseudo-identity generation and linkage with partial-private-key is executed by TRA and KGC in a sequential manner as follows:A vehicle, $V_{i}$, with its unique real identity denoted as $RID_{i}$ selects a random number $k_{i}\in Z_{q}^{*}$ and calculates $PID_{1}=k_{i}P$. Then the vehicle, $V_{i}$, sends $\left(RID_{i},PID_{1}\right)$ to the TRA through a secure channel.The TRA first checks the $RID_{i}$, if its acceptable then it calculates, $PID_{2}=RID_{i}⊕H_{1}(\beta .PID_{1}\left|\right|T_{i}\left|\right|T_{pub})$, where $T_{i}$ indicates the validity period of the pseudo-identity. The pseudo-identity that is used to identify a vehicle, $V_{i}$, is $ID_{i}=(PID_{1}||PID_{2}\left|\right|T_{i})$ and it is sent to the vehicle and KGC through a secure channel. During revocation TRA obtains the real identity by computing $RID_{i}=PID_{2}⊕H_{1}\left(\beta |\right|T_{i}\left|\right|T_{pub})$.Upon receipt of the pseudo-identity, $ID_{i}$, KGC chooses a random number, $d_{i}\in Z_{q}^{*}$ and computes $Q_{ID_{i}}=d_{i}P$ and then computes the partial private key, $psk_{i}$, for the vehicle, $V_{i}$, as $psk_{i}=d_{i}+H_{2}(ID_{i}\left|\right|Q_{ID_{i}})\times \alpha \;mod\;p$.The KGC then sends the pseudo-identity and partial private key $\left(Q_{ID_{i}},psk_{i}\right)$ to the vehicle, $V_{i}$, through a secure channel.The vehicle is able to check the authenticity of the pseudo-identity and the partial private key received from the KGC by verifying whether $psk_{i}.P=Q_{ID_{i}}+H_{2}(ID_{i}\left|\right|Q_{ID_{i}}).P_{pub}$. The conditional privacy-preservation is enhanced in the design by combining the secret contribution from the vehicle, $V_{i}$, itself and the TRA on the other hand. It is designed in such a way that the TRA is able to revoke the real identity of the vehicle when needed to do so. At the end of it all, the pseudo-identity and the partial private key are stored in the tamper-proof devices in the vehicle.3.
*Vehicle-Key-Generation*
The vehicle, $V_{i}$, randomly selects a secret value $x_{i}\in Z_{q}^{*}$ as its secret key noted as $vsk_{i}$ and then calculates its corresponding public key $vpk_{i}=x_{i}.P$. Then $V_{i}$ set the full private key as $sk_{i}=x_{i}+psk_{i}$.4.
*Sign*
The message signature is necessary for the sake of upholding the authentication and integrity of the message to the receiver of the message who rightly does verification. The vehicle, $V_{i}$, selects one of its stored pseudo-identity, $ID_{i}$, and picks the latest timestamp, $t_{i}$. With the signing Keys $\left(psk_{i},sk_{i}\right)$ and the traffic related message $M_{i}$, the vehicle $V_{i}$ carries out the following steps to produce a signature.Selects a random number $r_{i}\in Z_{q}^{*}$ and computes $R_{i}=r_{i}P$.Computes,
(1)$$h_{i}=H_{3}(M_{i}||ID_{i}\left|\right|Q_{ID_{i}}||vpk_{i}\left|\right|R_{i}\left|\right|t_{i})$$
and
(2)$$S_{i}=h_{i}.r_{i}+sk_{i}modp,$$
then, $V_{i}$ computes,
(3)$$\sigma_{i}=\left(R_{i},S_{i}\right)$$Here $\sigma_{i}$, is the computed certificate-less signature on the traffic related data $M_{i}$ for latest timestamp $t_{i}$ and identification $ID_{i}$.Then the final message that, $V_{i}$ sends to nearby RSU and vehicles for verification is $\left(ID_{i},Q_{ID_{i}},vpk_{i},M_{i},t_{i},\sigma_{i}\right)$.These steps are routinely carried out every time, $V_{i}$ sends a message to RSU.5.
*Individual Verify*
On receipt of the certificate-less signature $\sigma_{i}=\left(R_{i},S_{i}\right)$ on the traffic related data $M_{i}$ and timestamped at $t_{i}$ signed by the vehicle along with its public key $vpk_{i}$, if the received $T_{i}$ in $ID_{i}$ and $t_{i}$ are both valid, then the RSU performs the following procedures.Computes
(4)$$h_{i,0}=H_{2}(ID_{i}\left|\right|Q_{ID_{i}})$$
and
(5)$$h_{i}=H_{3}(M_{i}||ID_{i}\left|\right|Q_{ID_{i}}||vpk_{i}\left|\right|R_{i}\left|\right|t_{i})$$Verifies whether
(6)$$S_{i}.P=h_{i}.R_{i}+vpk_{i}+Q_{ID_{i}}+h_{i,0}.P_{pub},$$
holds or not.The RSU accepts the certificate-less signature if the verification holds. Correctness checking works, since $P_{pub}=\alpha .P$, $Q_{ID_{i}}=d_{i}.P$, $psk_{i}=d_{i}+H_{2}(ID_{i}\left|\right|Q_{ID_{i}})\times \alpha \;modp$, $R_{i}=r_{i}.P$, $sk_{i}=x_{i}+psk_{i}$, $h_{i,0}=H_{2}(ID_{i}\left|\right|Q_{ID_{i}})$ and $S_{i}=h_{i}.r_{i}+sk_{i}\;modp$. Thus the computation proceeds as follows:
$$\begin{array}{cc} S_{i}.P & =(h_{i}.r_{i}+sk_{i}).P\\ \; & =h_{i}.r_{i}.P+\left(x_{i}+psk_{i}\right)P\\ \; & =h_{i}.R_{i}+x_{i}.P+psk_{i}.P\\ \; & =h_{i}.R_{i}+vpk_{i}+[d_{i}+H_{2}(ID_{i}\left|\right|Q_{ID_{i}})\alpha ]\\ \; & =h_{i}.R_{i}+vpk_{i}+Q_{ID_{i}}+\left(h_{i,0}.\alpha \right)P\\ \; & =h_{i}.R_{i}+vpk_{i}+Q_{ID_{i}}+h_{i,0}.P_{pub}. \end{array}$$However, for purposes of saving computation cost, it is recommended to do data aggregation and batch verification on the signatures from the network environment of a particular RSU.6.
*Aggregate*
Each RSU is an out-posted aggregate signature generator that collects individual certificate-less signatures into a single verifiable one. The components come from an aggregating set *V* on *n* vehicles, $\left\{V_{1},V_{2},\cdots ,V_{n}\right\}$ whose corresponding pseudo-identities are $\left\{ID_{1},ID_{2},\cdots ,ID_{n}\right\}$ with public keys $\left\{vpk,vpk_{2},\cdots ,vpk_{n}\right\}$ and message signature pairs $\left(M_{1},t_{1},\sigma_{1}\right)$, $\left(M_{2},t_{2},\sigma_{2}\right)$, ⋯, $\left(M_{n},t_{n},\sigma_{n}\right)$, where $\sigma_{i}=\left(R_{i},S_{i}\right)$ for i=1,2,⋯,n. The RSU or an application server for the traffic control center for instance computes the sum $S=\sum_{i=i}^{n}S_{i}$ and output an aggregate certificate-less signature as,
(7)$$\sigma =\left(R_{1},S_{1}\right),\left(R_{2},S_{2}\right),\cdots ,\left(R_{n},S_{n}\right),$$
for i=1,2,⋯,n.7.
*Aggregate Verify*
On receipt of the certificate-less aggregate signature σ from *n* vehicle $\left\{V_{1},V_{2},\cdots ,V_{n}\right\}$ whose pseudo-identities are $\left\{ID_{1},ID_{2},\cdots ,ID_{n}\right\}$ with corresponding public keys, $\left\{vpk,vpk_{2},\cdots ,vpk_{n}\right\}$ and the traffic related messages $\left\{M_{1}||t_{1},M_{2}||t_{2},\cdots ,M_{n}||t_{n}\right\}$ then the RSU or the application server carries out the following procedures, if both $T_{i}$ in $ID_{i}$ and $t_{i}$ are checked to be valid.RSU computes
(8)$$h_{i,0}=H_{2}(ID_{i}\left|\right|Q_{ID_{i}})$$
and
(9)$$h_{i}=H_{3}(M_{i}||ID_{i}||vpk_{i}\left|\right|R_{i}\left|\right|t_{i})$$
for i=1,2,⋯,nRSU verifies if the computation holds,
(10)$$S.P=\sum_{i=i}^{n}h_{i}.R_{i}+\sum_{i=i}^{n}vpk_{i}+\sum_{i=i}^{n}Q_{ID_{i}}+\sum_{i=i}^{n}h_{i,0}.P_{pub}.$$If the verification holds, then the RSU accepts the aggregate certificate-less signature. The computation is valid by the correctness check, since $P_{pub}=\alpha .P$, $Q_{ID_{i}}=d_{i}P$, $psk_{i}=d_{i}+H_{2}(ID_{i}\left|\right|Q_{ID_{i}})\times \;modp$, $R_{i}+r_{i}P$, $S_{i}=h_{i}.r_{i}+psk_{i}\;modp$, and $S=\sum_{i=i}^{n}S_{i}$, thus we obtain.
$$\begin{array}{cc} S_{i}.P & =\sum_{i=i}^{n}\left(h_{i}.r_{i}+sk_{i}\right).P\\ \; & =\sum_{i=i}^{n}h_{i}.r_{i}.P+\sum_{i=i}^{n}\left(x_{i}+psk_{i}\right)P\\ \; & =\sum_{i=i}^{n}h_{i}.R_{i}+\sum_{i=i}^{n}x_{i}.P+\sum_{i=i}^{n}psk_{i}.P\\ \; & =\sum_{i=i}^{n}h_{i}.R_{i}+\sum_{i=i}^{n}vpk_{i}+\sum_{i=i}^{n}[d_{i}+H_{2}(ID_{i}\left|\right|Q_{ID_{i}})\alpha ]P\\ \; & =\sum_{i=i}^{n}h_{i}.R_{i}+\sum_{i=i}^{n}vpk_{i}+\sum_{i=i}^{n}Q_{ID_{i}}+\sum_{i=i}^{n}\left(h_{i,0}.\alpha \right)P\\ \; & =\sum_{i=i}^{n}h_{i}.R_{i}+\sum_{i=i}^{n}vpk_{i}+\sum_{i=i}^{n}Q_{ID_{i}}+\sum_{i=i}^{n}h_{i,0}.P_{pub}. \end{array}$$

## 5. Analyses

From here on, we will devote to giving a formal security proof, security privacy preservation analyses and then we will present the performance evaluation of the proposed ECLAS scheme with conditional privacy-preservation for a VANETs enhanced smart grid.

### 5.1. Security Proof

In this section now, we will provide security proof for the proposed ECLAS scheme for VANETs. We assume the security model for CLAS schemes where there are two types of adversaries, which are *Type 1 Adversary* and *Type 2 Adversary* as demonstrated in the security model for CLAS scheme.

**Theorem** **1.***Under the assumption that ECDL in G is intractable, then the proposed scheme*$\left(\epsilon ,t,q_{c},q_{s},q_{h}\right)$, *is secure against adversary 1 in random oracle model, where*$q_{c},q_{s},q_{h}$ are the ***Create***, ***Sign** and **Hash** queries respectively which the adversary is allowed to make*.

**Proof.** Suppose there is a probabilistic polynomial time adversary $A_{1}$, we construct an algorithm F that solves the ECDL problem by utilizing $A_{1}$. Assume that F is given an ECDL problem instance, (P,Q) to compute $x\in Z_{q}^{*}$ so that Q=xP. Thus, F chooses a challenging identity $ID^{*}$ for the identity ID to answer any random queries from $A_{1}$ as follows:
**Set-up (ID) Query**: The challenger F selects its random numbers $\alpha^{*}$ and $\beta^{*}$ as its master keys and has a corresponding public key as $P_{pub}^{*}=\alpha^{*}P$ and $T_{pub}^{*}=\beta^{*}P$ then sends the system parameters $\left\{P,p,q,E,G,H_{2},H_{3},P_{pub}^{*},T_{pub}^{*}\right\}$ to $A_{1}$.**Create (ID) Query**: F stores the hash list $L_{C}$ of the tuple $\left(ID,Q_{ID_{i}},vpk_{i},psk_{i},sk_{i},h_{2}\right)$. Whenever an adversary $A_{1}$ makes a query for ID, and if the ID is contained in $L_{C}$, then F returns $\left(ID,Q_{ID_{i}},vpk_{i},psk_{i},sk_{i},h_{2}\right)$ to $A_{1}$. Then F, execute the oracle as follows. if $ID=ID^{*}$, F randomly chooses the values $a,b,c\in Z_{q}^{*}$ and sets $Q_{ID}=a.P_{pub}^{*}+b.P$, $vpk_{i}=c.P$, $psk_{i}=b$, $sk_{i}=c$, $h_{2}=H_{2}\left(ID|\right|Q_{ID})\leftarrow amodq$, then F adds $\left(ID,Q_{ID},h_{2}\right)$ to the list $L_{H_{2}}$ and returns $\left(ID,Q_{ID_{i}},vpk_{i},psk_{i},sk_{i},h_{2}\right)$ to $A_{1}$. as the equation $psk_{i}.P=Q_{ID}+h_{2}.P_{pub}^{*}$, thereby implying that the partial private key is valid.**$H_{2}$ Query**: Whenever an $H_{2}$ query with $\left(ID,Q_{ID}\right)$ is made, and ID is already in the hash list $L_{H_{2}}$, then F reply with a corresponding $h_{2}$. On the other hand, F runs Create(ID) to obtain $h_{2}$ and then sends $h_{2}$ to $A_{1}$.**Partial-Private-Key-Extract (ID) Query**: If $ID^{*}=ID$, then F aborts the game. Otherwise, F looks in the hash list $L_{C}$, if ID is found in the list, then F returns $psk_{i}$ to $A_{1}$. If ID is not in the list $L_{C}$, F executes Create(ID) query to obtain $psk_{i}$ and sends it to $A_{1}$.**Public-Key (ID) Query**: Upon receiving the query on ID, when ID is already in the list $L_{C}$, F replies with $pk=(Q_{ID},vpk_{i})$. On the other hand, F executes Create(ID) query to obtain $\left(Q_{ID},vpk_{i}\right)$ and sends it to $A_{1}$.**Public-Key-replacement $(ID,pk^{\prime }$) Query**: F stores the hash list $L_{R}$ of tuple $\left(ID,d_{i},Q_{ID},sk_{i},vpk_{i}\right)$. When $A_{1}$ executes the query with $\left(ID,pk^{\prime }\right)$, where $Q_{ID}^{\prime }=d^{\prime }.P$, $vpk_{i}^{\prime }=x_{i}^{\prime }.P$ and $pk\prime =(Q_{ID}^{\prime },vpk_{i}^{\prime })$, then F sets $Q_{ID}=Q_{ID}^{\prime }$, $vpk_{i}=vpk_{i}^{\prime }$, $psk_{i}=⊥$ and $x_{i}=x_{i}^{\prime }$. Then the challenger F, updates the list $L_{R}$ to be $\left(ID,d_{i}^{\prime },Q_{ID}^{\prime },vpk_{i}^{\prime },x_{i}^{\prime }\right)$.**$H_{3}\left(ID\right)$ Query**: F keeps the hash list $L_{H_{3}}$ of the tuple $\left(m,ID,R,vpk_{i},t,h_{3}\right)$ and if the ID queries are not in the list, F replies with $h_{3}$. Otherwise, it selects a random number $h_{3}$ such that $h_{3}=H_{3}(m||ID||vpk_{i}\left||R||t\right)$ then add it to the list $L_{H_{3}}$ and returns $h_{3}$ to $A_{1}$**Sign (ID,m) Query**: $A_{1}$ makes a sign query on (ID,m), once ID is on the list $L_{R}$, F chooses random numbers $a,b,c\in Z_{q}^{*}$, and sets s=a, R=P, $h_{3}=H_{3}(m||ID||vpk_{i}\left||R||t\right)\leftarrow \left(a-b-c\right)modq$ and then inserts $\left(m,ID,R,vpk_{i},t,h_{3}\right)$ to the list $L_{H_{3}}$. The resultant signature is (R,s), and if ID is not in the list $L_{R}$, then F acts according to scheme’s procedure.□

As a result, $A_{1}$ produces a forged signature $\sigma =(R,s_{\left\{1\right\}})$ on the message (ID,m) which passes verification process. If $ID\neq ID^{*}$, F aborts the process. F keeps on challenging $A_{1}$ up until it responds to the $H_{3}$ query. $A_{1}$ will be prompted to generate another valid signature $\sigma =(R,s_{\left\{2\right\}})$ by using the same *R*. Thus we have:(11)$$s_{\left\{i\right\}}.P=h_{3\left\{i\right\}}.R+vpk_{i}+Q_{ID}+h_{2}.P_{pub}^{*},$$
where i=1,2.

By solving the two linear equations we obtain the value of *r* by
(12)$$\frac{s_{2}-s_{1}}{h_{\left\{2\right\}}-h_{\left\{1\right\}}},$$
similarly, with continuous querying, $H_{2}$ will allow computation of *x*.

**Probabilistic Analysis**: The simulation of Create(ID) queries fails when the random oracle assignment $H_{2}\left(ID|\right|Q_{ID})$ causes inconsistency with the probability of at most $\frac{q_{h}}{q}$. The probability of successful simulation of $q_{c}$ times is at least ${\left(1-\frac{q_{h}}{q}\right)}^{q_{c}}\geq 1-\left(\frac{q_{h}q_{c}}{q}\right)$. Similarly, the simulation is $q_{h}$ successful with the probability of at least ${\left(1-\frac{q_{h}}{q}\right)}^{q_{h}}\geq \left(1-\frac{q_{h}^{2}}{q}\right)$ and $ID=ID^{*}$ with the probability of $\frac{1}{q_{c}}$. Thus, in overall the probability of successful simulation is
(13)$$\left(1-\frac{q_{h}q_{c}}{q}\right)\left(1-\frac{q_{h}^{2}}{q}\right)\left(\frac{1}{q_{c}}\right)\epsilon .$$

**Theorem** **2.**
*Under the assumption that ECDL in G is intractable, then the proposed scheme $\left(\epsilon ,t,q_{c},q_{s},q_{h}\right)$, is secure against adversary 2 in random oracle model, where $q_{c},q_{s},q_{h}$ are the*
***Create***
*,*
***Sign***
*and*
***Hash***
*queries respectively which the adversary is allowed to make.*


**Proof.** Suppose there is a probabilistic polynomial time adversary $A_{2}$, we construct an algorithm F that solves the ECDL problem by utilizing $A_{2}$. Assume that F is given a ECCDH problem instance, (P,Q) to compute $x,y\in Z_{q}^{*}$ so that Q=xyP. Thus, F chooses an challenging identity $ID^{*}$ for the identity ID to answer any random queries from $A_{2}$ as follows:
**Set-up (ID) Query**: The challenger F selects its random numbers $\alpha^{*}$ and $\beta^{*}$ as its master keys and has a corresponding public key as $P_{pub}^{*}=\alpha^{*}P$ and $T_{pub}^{*}=\beta^{*}P$ then sends the system parameters $\left\{P,p,q,E,G,H_{2},H_{3},P_{pub}^{*},T_{pub}^{*}\right\}$ to $A_{2}$.**Create (ID) Query**: F stores the hash list $L_{C}$ of the tuple $\left(ID,Q_{ID_{i}},vpk_{i},psk_{i},sk_{i},h_{2}\right)$. Whenever an adversary $A_{2}$ makes a query for ID, and if the ID is contained in $L_{C}$, then F returns $\left(ID,Q_{ID_{i}},vpk_{i},psk_{i},sk_{i},h_{2}\right)$ to $A_{2}$. If $ID=ID^{*}$, F randomly selects $a,b\in Z_{q}^{*}$ and computes $Q_{ID}=aP$, $vpk_{i}=Q$, $h_{2}=H_{2}\left(ID|\right|Q_{ID})\leftarrow b$, $psk_{i}=a+x.h_{2}$, $sk_{i}=⊥$. If $ID=\neq ID^{*}$, F, randomly selects $a,b,c\in Z_{q}^{*}$ and computes $Q_{ID}=a.P$, $vpk_{i}=b.P$, $h_{2}=H_{2}\left(ID|\right|Q_{ID})\leftarrow c$, $psk_{i}=a+x.h_{2}$, $sk_{i}=b$. Then F, responds to the query with $\left(ID,Q_{ID_{i}},vpk_{i},psk_{i},sk_{i},h_{2}\right)$ and then appends $\left(ID,Q_{ID},h_{2}\right)$ to the hash list $L_{H_{2}}$.**$H_{2}$ Query**: Whenever an adversary $A_{2}$ makes an $H_{2}$ query with $\left(ID,Q_{ID}\right)$, and ID is already in the hash list $L_{H_{2}}$, then F reply with a corresponding $h_{2}$. On the other hand, F runs Create(ID) to obtain $h_{2}$ and then sends $h_{2}$ to $A_{2}$.**Partial-Private-Key-Extract (ID) Query**: Upon receipt of the query on ID, F verifies from the hash list $L_{C}$, if ID is found to be in the hash list F returns $psk_{i}$ to $A_{2}$. If ID is not in the hash list, $L_{C}$, F executes Create(ID) query to obtain $psk_{i}$ and sends it to $A_{2}$.**Public-Key (ID) Query**: Upon receipt of query on ID, when ID is already in the list $L_{C}$, F replies with $pk=(Q_{ID},vpk_{i})$. On the other hand, F executes Create(ID) query to obtain $\left(Q_{ID},vpk_{i}\right)$ and sends it to $A_{2}$.**Secret-Key-Extract (ID) Query**: On receipt of the queries from $A_{2}$, if $ID=ID^{*}$, F stops the simulation. While, if ID is already in the list $L_{C}$, then F reply with $sk_{i}$. Whereas if, ID is not in the list $L_{C}$, F executes Create(ID) query to obtain $\left(ID,Q_{ID},vpk_{i},psk_{i},sk_{i},h_{2}\right)$ and sends $sk_{i}$ to $A_{2}$.**$H_{3}\left(ID\right)$ Query**: F keeps the hash list $L_{H_{3}}$ of the tuple $\left(m,ID,R,vpk_{i},t,h_{3}\right)$ and if the ID queries are in the list, F replies with $h_{3}$. Otherwise, it selects a random number $h_{3}$ such that $h_{3}=H_{3}(m||ID||vpk_{i}\left||R||t\right)$ then add it to the list $L_{H_{3}}$ and returns $h_{3}$ to $A_{2}$**Sign (ID,m) Query**: As $A_{2}$ makes a sign query on (ID,m), once $ID\neq ID^{*}$, F acts according to protocol flow. Otherwise, F randomly chooses the values $a,b,f\in Z_{q}^{*}$ and sets s=a, $h_{3}=H_{3}(m||ID||vpk_{i}\left||R||t\right)\leftarrow f$, $R=h_{3}^{-1}\left(bP_{pub}^{*}-Q\right)$, and returns the signature (R,s). If the verification, $s.P=h_{3}.R+Q_{ID}+vpk_{i}+h_{2}.P_{pub}^{*}$, holds then the signature is valid.
□

As a result, $A_{2}$ produces a forged signature $\sigma =(R,s_{\left\{2\right\}})$ on the message (ID,m) which passes verification process. If $ID\neq ID^{*}$, F aborts the process. F keeps on challenging $A_{2}$ up until it responds to the $H_{3}$ query. $A_{2}$ will be prompted to generate another valid signature $\sigma =(R,s_{\left\{2\right\}})$ by using the same *R*. Thus we have:(14)$$s_{\left\{i\right\}}.P=h_{3\left\{i\right\}}.R+vpk_{i}+Q_{ID}+h_{2}.P_{pub}^{*},$$
(15)$$s_{\left\{i\right\}}=h_{3\left\{i\right\}}.r+y+d_{i}+h_{2}.x,$$
where i=1,2.

By solving the two linear equations involving *r* and *y* as variables, we can derive the value of *y* as an output of ECDL problem.

### 5.2. Security and Privacy-Preservation Analyses

This part discusses the security and privacy-preservation features satisfied by the proposed scheme, specifically this is in respect to anonymity (identity privacy), message authentication, data integrity, traceability, unlinkability and resistance to attacks.

Anonymity: In the proposed scheme the vehicle’s identification $ID_{i}$ is not the real identification $RID_{i}$, but rather a pseudo-identity as offered by the TRA for purposes of achieving conditional privacy of the vehicle in VANETs. The only way for an adversary or any malicious party to obtain the real identity it by computing $RID_{i}=ID_{i}⊕H_{1}(\beta .PID_{1}\left|\right|T_{i}\left|\right|T_{pub})$. Without knownledge of the TRA’s master private key β, no other party can know the vehicle’s real identity $RID_{i}$, since it requires β to calculate $H_{1}(\beta .PID_{1}\left|\right|T_{i}\left|\right|T_{pub})$. This manipulation is infeasible for an adversary to achieve since the extraction of β from $T_{pub}=\beta .P$, involves an intractable ECDL problem. Therefore, these claims ascertain the satisfaction of user identity privacy-preservation.Message Integrity and Authentication: By virtue of signing a message before broadcasting, the legitimate user’s authenticity is verified. Based on the ECDLP assumption the authenticity and integrity of the message $\left(ID_{i},Q_{ID_{i}},vpk_{i},M_{i},t_{i},\sigma_{i}\right)$ is upheld by verifying the computation $S_{i}.P=h_{i}.R_{i}+vpk_{i}+Q_{ID_{i}}+h_{i,0}.P_{pub}$. Since $h_{i}=H_{3}(M_{i}||ID_{i}\left|\right|Q_{ID_{i}}||vpk_{i}\left|\right|R_{i}\left|\right|t_{i})$ and $h_{i,0}=H_{2}(ID_{i}\left|\right|Q_{ID_{i}})$, no malicious party can forge $\sigma_{i}=\left(R_{i},S_{i}\right)$ which achieves the maessage integrity and authentication of which needs knoweledge of full private key $sk_{i}=x_{i}+psk_{i}$ in its formulation.Traceability: Although the vehicle is identified by a pseudonym, in necessary circumstances the real identity of a particular vehicle can be mapped back from the pseudonym. For instance, the pseudo-identity of a vehicle is $ID_{i}=(PID_{1}||PID_{2}\left|\right|T_{i})$ and the TRA can revoke the real identity by calculating $PID_{2}=$$RID_{i}⊕H_{1}(\beta .PID_{1}\left|\right|T_{i}\left|\right|T_{pub})$. As such, once a vehicle is flagged as questionable the TRA is able to trace its true identity and thereby carrying out whatever necessary procedures to curb any kind of malpractice. Once this is done the TRA records the real identity $RID_{i}$ on the revocation list of the system and as a result the vehicle cannot use its corresponding pseudo-identity $ID_{i}$.Unlinkability: The message transmitted $\left(ID_{i},Q_{ID_{i}},vpk_{i},M_{i},t_{i},\sigma_{i}\right)$ from a vehicle $V_{i}$ to others has the component $PID_{1}=k_{i}P$, where $k_{i}\in Z_{q}^{*}$ is random, that is randomly generated for any particular message transmitted. Since the $PID_{1}$ is also a component for pseudo-identity generation, it means the randomness in $PID_{1}$ results in the randomness of the publicized pseudo-identity $ID_{i}$, hence, any two individual captures of the pseudo-identity $ID_{i}$ for $V_{i}$ stills seem random and unrelated to the real identity $RID_{i}$, in the eyes of eavesdroppers. So by virtue of the identification being anonymous and distinct any captured signatures cannot be linked to previously captured identity nor to a particular true signer. Thus, any communication is seen as random and new in the plying eyes of an adversary and has no any relationship to previous communications for an eavesdropper to learn any useful information from such communication.Resistance to Attacks: At this point we will present a demonstration of how the proposed ECLAS scheme can resist the main common attacks such as—replay attack, modification attack, impersonation attack, and stolen verifier attack.–Replay Attack Resilience: In the message $\left(ID_{i},Q_{ID_{i}},vpk_{i},M_{i},t_{i},\sigma_{i}\right)$ the $t_{i}$ in the message helps in checking replay attacks. The recipients, RSUs or vehicles will have to check the freshness of the message, and once the timestamp is invalid the message is discarded. As such the proposed scheme, ECLAS, could resist against replay attack.–Modification Attack Resilience: In the scheme a valid message $\left(ID_{i},Q_{ID_{i}},vpk_{i},M_{i},t_{i},\sigma_{i}\right)$ has a valid digital conditional anonymous signature $\left(ID_{i},\sigma_{i}\right)$. Any modification to the message $\left(ID_{i},Q_{ID_{i}},vpk_{i},M_{i},t_{i},\sigma_{i}\right)$ can be detected during verification $S_{i}.P=h_{i}.R_{i}+vpk_{i}+Q_{ID_{i}}+h_{i,0}.P_{pub}$ which simultaneously authenticates the sender, $V_{i}$, and the TA side of TRA and KGC. Therefore, the proposed ECLAS scheme stands against modification attack.–Impersonation Attack Resilience: It is not feasible for an attacker to launch a successful impersonation on the message $\left(ID_{i},Q_{ID_{i}},vpk_{i},M_{i},t_{i},\sigma_{i}\right)$ of which can pass verification as if it was generated by a legal user $V_{i}$. However, it is impossible for an attacker to obtain the KGC’s master key α and the users private key $x_{i}$ from the publicly accessible parameters as it will involve solving the intractable problems of ECDLP and ECCDHP from $vpk_{i}=x_{i}P$ and $P_{pub}=\alpha P$.–Stolen Verifier Table Attack Resilience: In the proposed ECLAS scheme, both the TA side, which comprises of TRA and KGC and the user side, which comprises of RSUs and OBUs on the vehicle do not require a check list. This implies resistance against stolen verification table attack as it means the table can not be stolen.–Key-Escrow Resilience: Although the TAs side has access to the master keys used for generating the user’s partial private key, still more neither TRA nor KGC can generate a valid signature $\sigma_{i}=\left(R_{i},Si\right)$ for a valid message $\left(ID_{i},Q_{ID_{i}},vpk_{i},M_{i},t_{i},\sigma_{i}\right)$. This is due to the fact that, the vehicle adds a secret value $x_{i}$ to the partial private key $psk_{i}$ when computing its full private key $sk_{i}=x_{i}+d_{i}+H_{2}(ID_{i}\left|\right|Q_{ID_{i}})\alpha$, which is used for signing messages. To this effect although TRA knows the master key β and KGC knows the master key α for the systems, they cannot forge messages to masquerade as $V_{i}$ illegally. Thus, the proposed ECLAS scheme withstands the key escrow attacks.

Now we will present a comparison analysis of ECLAS with recent related works in terms of security features satisfied. In Table 2 the results of the comparison is provided with the features coded as, SF-1, SF-2, SF-3, SF-4, SF-5, SF-6 to denote, integrity and authentication, anonymity, traceability and revocability, unlinkability, key escrow problem and resistance to common attacks respectively. In the Table 2 the symbol denotes the satisfaction whereas ✗, denotes not satisfaction of the security feature. As shown by the comparison table, the schemes in [47,53,54] fall short from fulfilling some of the features.

### 5.3. Performance Evaluation

In this section, we will present the performance analysis of the proposed ECLAS scheme in terms of comparable feature with related research on the fields that gives merit to the proposed scheme. As such, performance comparison features are discussed in terms of computation cost analysis and communication cost analysis. We will assess the performance evaluation of the proposed work in terms of computation cost comparison against other related works by adopting the method presented in [17]. In [17] bilinear pairing on an 80 bits security parameter length is created as $:G_{1}\times G_{2}\to G_{T}$. Here we consider $G_{1}$ as an additive group of order *q* defined on a super-singular elliptic curve $E:y^{2}=x^{3}+xmodp$ of embedding degree of 2. The recommended security parameter length for *q* and solinas prime number *p* are taken as 512 bits and 160 bits, respectively.

For convenience, we will define the notations for execution time for different cryptographic computations in the schemes under discussion as portrayed in Table 3. We borrow the execution times directly from [17], which was evaluated using the MIRACL cryptographic library, to assess the efficiency of schemes. Operations which are very light like addition operation in $Z_{q}^{*}$ and the multiplication operation in $Z_{q}^{*}$ will not be considered.

The notation for various computation operations are as follows.

$T_{bp}$: Denotes execution time for bilinear pairing operation defined as, e(P,Q), where $P,Q\in G_{1}$

$T_{bp.m}$: Denotes execution time for scalar multiplication operation x.P, that is related to pairing operation defined as e(P,Q), where $P,Q\in G_{1}$, and $x\in Z_{q}^{*}$

$T_{bp.sm}$: Denotes execution time for small scalar multiplication operation, $v_{i}.P$, that is related to pairing operation e(P,Q), where $P,Q\in G_{1}$ and $v_{i}\in \left[1,2^{t}\right]$ is a small random integer, for a small predefined integer *t*.

$T_{bp.a}$: Denotes execution time for point addition in bilinear pairing operation e(P,Q), such that R=P+Q, where $R,P,Q\in G_{1}$

$T_{H}$: Denotes execution time for map-to-point hash function operation related to pairing operation e(P,Q), where $P,Q\in G_{1}$.

$T_{e.m}$: Denotes execution time for scalar multiplication operation, x.P, over ECC group, where P∈G and $x\in Z_{q}^{*}$.

$T_{e.sm}$: Denotes execution time for small scalar multiplication operation, $v_{i}.P$, for small exponent test, where P∈G and $v_{i}\in \left[1,2^{t}\right]$ is a small random integer, for a small predefined integer *t*.

$T_{e.a}$: Denotes execution time for point addition operation over an elliptic curve group, R=P+Q, where R,P,Q∈G.

$T_{h}$: Denotes execution time for one hash function operation.

#### 5.3.1. Computation Cost Analysis

In this section, we give a formal security proof on the proposed certificate-less signature scheme. While using the computation execution times for various dominant time-consuming cryptographic operations summarized in Table 3, we carry out a computation analysis of related CLAS schemes [2,13,23,27,55] in terms of the three phases of message signing, individual verify and aggregate verify overhead in RSU. The observation is clear that our proposed scheme, ECLAS, has better computation performance to related works from Table 4. In [27], to generate a signature a vehicle carries out three scalar multiplication, $3T_{e.m}$, over an elliptic curve. This means the computation cost for signing is $3T_{e.m}\approx 1.326\;ms$. Whilst for verifying a signature, three bilinear pairings, one scalar multiplication over an elliptic curve and one map-to-point hash function operations, are required. Thus, individual verification needs $2T_{bp}+T_{e.m}+T_{H}\approx 17.481\;ms$. In aggregate verification phase, three bilinear pairings, *n* scalar multiplication over elliptic curve and *n* map-to-point hash function operations are required, $2T_{bp}+nT_{e.m}+nT_{H}\approx 12.633+4.4198n\;ms$. In the proposed ECLAS scheme, for signature generation a vehicle requires two scalar multiplication with respect to elliptic curve and one hash function operation, $2T_{e.m}+T_{h}$, amounting to the computation load of $2T_{e.m}+T_{h}\approx 0.8841\;ms$. For individual signature verification, ECLAS, similarly requires two scalar multiplication with respect to elliptic curve and one hash function operation, $2T_{e.m}+T_{h}$, amounting to the computation load of $2T_{e.m}+T_{h}\approx 0.8841\;ms$. Whereas for aggregate signature verification, ECLAS requires 2n scalar multiplication with respect to elliptic curve and *n* hash function operation, $2nT_{e.m}+nT_{h}$, yielding computation cost of $2nT_{e.m}+nT_{h}\approx 0.8841n\;ms$. in a similar manner, the computation cost for other relevant comparable schemes [2,13,23,55] can be calculated. Based on the generated summary results of computation cost comparison done in Table 4 and the visual representation done given in Figure 2 we make conclusion on the performance of ECLAS. It is clear that the proposed ECLAS scheme has all over computation efficiency compared to the rest of the scheme except [13], and although it has a slightly lower signing computation overhead it was found to have security flaws in [23], whereas the proposed scheme satisfies the security requirements and withstands KGC escrow property.

For simplicity sake, by regarding equal computation capabilities for signing and verifying then we can lump up the computation load that is incurred in message signing and individual verifying for a single signature. As such, the overall load for Horng et al. [27] comes up to (1.326+17.481)ms=18.807ms and for Cui et al. [13] the overall load is (0.4439+1.3298)ms=1.7737ms. Proceeding in this manner for the rest of the schemes, in Xiong et al. [55], Tzeng et al. [2], Kamil et al. [23] the overall computation loads are; 24.3675ms, 19.664ms, 2.1887ms respectively. Subsequently, ECLAS has an overall computation load of 1.7682ms, which is better than the rest as shown in Figure 2.

The relationship of verification time delay for particular number of aggregate signatures that RSU takes to compute for the schemes [2,13,23,27,55] is portrayed in the Figure 3.

As a requirement in VANETs, vehicles have to broadcast their messages every 100–300 ms, thus it entails that an RSU or AS can receive about 180 messages every 300ms. Therefore, in one second an RSU is expected to verify about 600–2000 messages [23]. In Figure 3, it endeavors to illustrate the time it takes to do batch verification for 2000 signatures. Thus, the comparative analysis shows that the proposed scheme has less verification time delay for *n* signature aggregation and the number of signatures has a direct proportion linear relationship to the verification delay.

#### 5.3.2. Communication Cost Analysis

In this part now, we will present the communication overhead of the proposed scheme against the related schemes [2,13,23,27,55] by borrowing experiment results from [17] to account for transmission cost for sending packets from vehicle to RSUs in V2I or V2V communication in VANETs, the sizes of elements in $G_{1}$ and *G* are 128 bytes and 40 bytes respectively. In addition, the elements in $Z_{q}^{*}$, the hash function value and timestamps are of the sizes 20 bytes, 20 bytes and 4 bytes respectively. We will consider the message traffic load for signatures only.

In [27], the vehicle broadcast the message $\left(ID_{i},vpk_{i},M_{i},t_{i},\sigma_{i}=\left(R_{i},S_{i}\right)\right)$ to RSUs, where $ID_{i},vpk_{i},R_{i},S_{i}\in G$ and $t_{i}$ is a timestamp. Therefore, the communication overhead is 3×40+4=124 bytes. In [13] the vehicle sends the message $\left(ID_{i},vpk_{i},Q_{ID_{i}},\sigma_{i}=\left(R_{i},S_{i}\right),t_{i}\right)$ to RSUs or AS, where $ID_{i},vpk_{i},Q_{ID_{i}},R_{i}\in G$, $S_{i}\in Z_{q}^{*}$ and $t_{i}$ is the timestamp. Thus, the communication load on the network is 4×40+20+4=184 bytes. In [55], the vehicle sends $\left(ID_{i},m_{i},upk_{i},signature\left(U_{i},V_{i}\right)\right)$ to RSU, which requires the bandwidth size of 4×40+20+4=184 bytes. Whereas, in [54] the message sent from a vehicle to RSU is $\left(PS_{j},PS1_{j},P_{i},PP_{i},\sigma_{i}=\left(U_{i},V_{ijk}\right)\right)$, where $PS_{j},PS1_{j},P_{i},PP_{i},U_{i},V_{ijk}\in G$. Therefore, the communication overhead is 6×128=768 bytes. In the proposed, ECLAS, scheme a vehicle sends traffic related signed message $\left(ID_{i},Q_{ID_{i}},vpk_{i},M_{i},t_{i},\sigma_{i}\right)$ to the verifier where $ID_{i}\in G$. Therefore, the total communication overhead is 4×40+20+4=184 bytes. The proposed scheme has less communication overhead load than [27,54] and is on a par with the schemes in [46,51,55] as outlined in Table 5.

However, these comparable works are found to be insecure in different aspects, like in [13], which so far has a decent efficient output, it was discovered that the scheme is insecure in [23,27].

## 6. Conclusions

In this paper, we presented an efficient certificate-less signature scheme with conditional privacy preservation for VANETs enhanced smart grid system that is based on elliptic curve cryptography and it provides user anonymity. The proposed work also removes the inherently key escrow problem associated with identity based cryptography by means of introducing a derivation of a full private key by the vehicle itself. Security proof under the random oracle model approach shows that the proposed scheme is secure by virtue of satisfying all the security requirements for VANETs. In this scheme certificate-less property is achieved without key escrow problem since the signature is derived by using a vehicle full private key which is not known by the KGC. Furthermore, the scheme does not require the computation intensive bilinear pairing and map-to-point hash function operations but rather is just based on less intensive operation over elliptic curve group in the design, hence achieving efficient computation cost. Even the communication overhead is within bounds with comparable schemes whilst achieving higher security merits. Thus, it is a comparatively efficient certificate-less aggregate signature scheme ideal for VANETs communications.

## Figures and Tables

**Figure 1 sensors-21-02900-f001:**
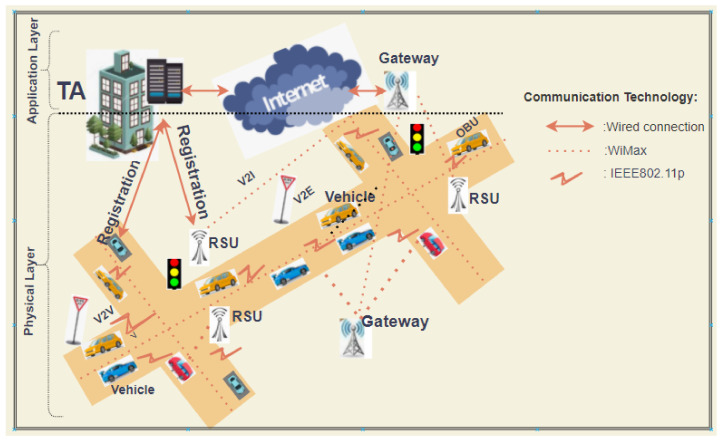
Two Layered Vehicular Ad hoc networks (VANETs) Architecture.

**Figure 2 sensors-21-02900-f002:**
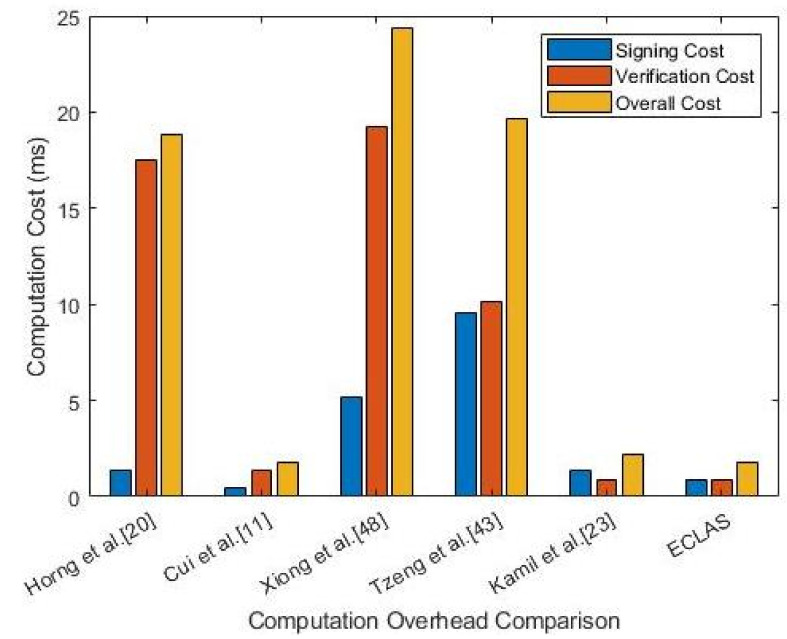
Computation Cost Comparison Per Unit.

**Figure 3 sensors-21-02900-f003:**
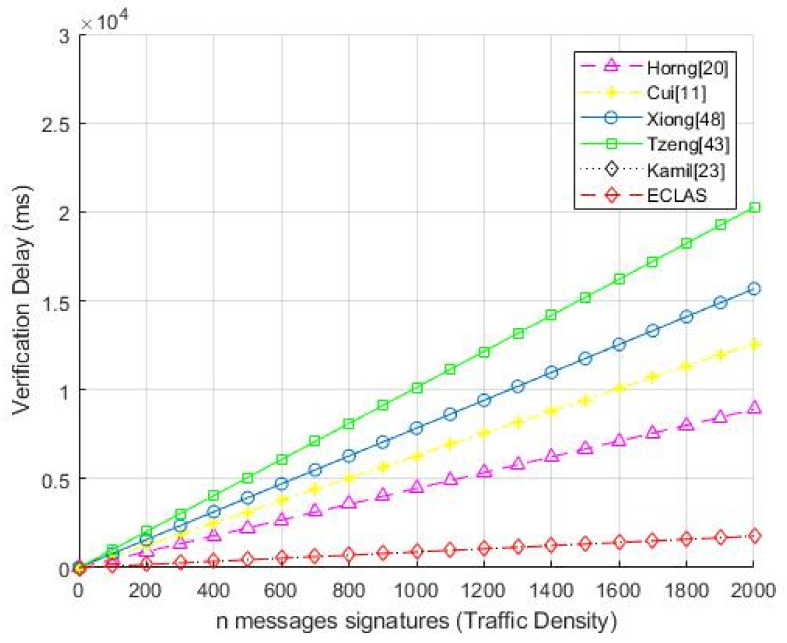
Verification Time Delays and Number of Signatures Relationship.

**Table 1 sensors-21-02900-t001:** Notations Used in the Proposed Scheme.

Symbols	Meanings of Symbols in the Scheme
$V_{i}$	$i^{th}$ vehicle
*p*, *q*	Two large primes
*E*	Is the chosen elliptic curve,
	$y^{3}=x^{2}+ax+bmodp$ where $a,b\in Z_{q}^{*}$
$E(F_{p})$	Is the prime field of an elliptic curve *E* order *p*
*P*	Is the generator of $E(F_{p})$ with large prime order *q*
*G*	A cyclic group generated by a point *P* on a non-singular
	elliptic curve *E*
$ID_{i}$	A pseudo-identity of $V_{i}$ such that $ID=(PID_{1},PID_{2},T_{i})$
$psk_{i}$	Partial private key for a vehicle, $V_{i}$
$\left(x_{i},x_{i}P\right)$	Secret key and public key for $V_{i}$
$sk_{i}$	Full private key for $V_{i}$
$T_{i}$	Validity period for the pseudo-identity $ID_{i}$ for $V_{i}$
$RID_{i}$	A real identity for the vehicle $V_{i}$
$\left(P_{pub},\alpha \right)$	KGC’s public key and master key respectively
$\left(T_{pub},\beta \right)$	TRA’s public key and master key respectively
$M_{i}$	Traffic-related message generated by $V_{i}$
$t_{i}$	Current timestamp
$H_{1}$, $H_{2}$, $H_{3}$	Hash function: $H_{1},\;H_{2}:{\left\{0,1\right\}}^{*}\to \;Z_{q}^{*}$
⊕	Exclusive-OR operation
||	concatenation

**Table 2 sensors-21-02900-t002:** Comparison Analysis of Security Features Satisfied.

Security	Alazzawi	Bayat	Malhi	ECLAS
Feature	et al. [47]	et al. [53]	et al [54]	
**SF-1**	✓	✓	✗	✓
**SF-2**	✓	✓	✓	✓
**SF-3**	✓	✓	✓	✓
**SF-4**	✗	✗	✓	✓
**SF-5**	✗	✗	✗	✓
**SF-6**	✓	✗	✗	✓

**Table 3 sensors-21-02900-t003:** Execution Times of Cryptographic Operations.

Operations	$T_{bp}$	$T_{\mathrm{bp}.m}$	$T_{\mathrm{bp}.\mathrm{sm}}$	$T_{\mathrm{bp}.a}$	$T_{H}$	$T_{e.m}$	$T_{e.\mathrm{sm}}$	$T_{e.a}$	$T_{h}$
**Times (ms)**	4.211	1.709	0.0535	0.0071	4.406	0.4420	0.0138	0.0018	0.0001

**Table 4 sensors-21-02900-t004:** Comparison of Computation Costs for Related Certificate-less aggregate signature (CLAS) Schemes in ms.

Schemes	Message Signing	Individual Verify	Aggregate Verify
Horng	$3T_{e.m}\approx 1.326\;ms$	$3T_{bp}+T_{e.m}+T_{H}$	$3T_{bp}+nT_{e.m}+nT_{H}$
et al. [27]		≈17.481ms	≈12.633+4.4198nms
Cui	$T_{e.m}+T_{e.a}+T_{h}$	$3T_{e.m}+2T_{e.a}+2T_{h}$	$\left(n+2\right)T_{e.m}+4nT_{e.a}$
et al. [13]	≈0.4439ms	≈1.3298ms	$+nT_{H}+nT_{h}$
			≈6.2973nms
Xiong	$3T_{bp.m}+2T_{bp.a}+T_{h}$	$3T_{bp}+2T_{bp.m}+T_{bp.a}$	$3T_{bp}+2nT_{bp.m}+nT_{bp.a}$
et al. [55]		$+T_{H}+T_{h}$	$+nT_{H}+nT_{h}$
	≈5.1413ms	≈19.2262ms	≈12.633+7.8312nms
Tzeng	$3T_{bp.m}+T_{H}$	$2T_{bp}+T_{bp.m}$	$2nT_{bp}+nT_{bp.m}$
et al. [2]			
	≈9.533ms	≈10.131ms	≈10.131nms
Kamil	$3T_{e.m}+2T_{e.a}+3T_{h}$	$2T_{e.m}+T_{e.a}+T_{h}$	$2nT_{e.m}+nT_{e.a}+nT_{h}$
et al. [23]	≈1.3297ms	≈0.8859ms	≈0.8859nms
ECLAS	$2T_{e.m}+T_{h}$	$2T_{e.m}+T_{h}$	$2nT_{e.m}+nT_{h}$
	≈0.8841ms	≈0.8841ms	≈0.8841nms

**Table 5 sensors-21-02900-t005:** Communication Overhead Summary.

Schemes	Sending of One Signature Message	Sending of *n* Signature Message
Horng et al. [27]	644 bytes	644n bytes
Cui et al. [13]	184 bytes	184n bytes
Xiong et al. [55]	184 btes	184n bytes
Malhi [54]	768 bytes	768n bytes
Kamil et al. [23]	184 bytes	184n bytes
ECLAS	184 bytes	184n bytes

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
