# Peer review of "Efficient Certificate-Less Aggregate Signature Scheme with Conditional Privacy-Preservation for Vehicular Ad Hoc Networks Enhanced Smart Grid System"

_sensors, 2021, doi:10.3390/s21092900_

Round 1

Reviewer 1 Report

The paper “Efficient Certificateless Aggregate Signature Scheme with Conditional Privacy-Preservation for Vecular Ad Hoc Networks Enhanced Smart Grid System” is presenting an interesting and complex problem for the future development of wireless communication technology of vehicles. However, I must make the following observations / comments:

  • In the title of the article a correction must be made, "Vehicular Ad Hoc Networks" instead of "Vecular Ad Hoc Networks".
  • The Figure 1 is not commented and discussed in the article’s text.
  • The phrase "Turing machine" (page 6, line 222) contains the name of a person, the English mathematician Alan Turing. Consequently, in the expression "Turing machine" the name Turing must be capitalized.
  • The Table 1 is not commented or discussed in the article’s text.
  • Table 3 is included in the text before its description, which is not acceptable. The same observation for Table 4 and Table 5.
  • The authors use very long sentences, the meaning of which is difficult to understand. For example: "The computation cost for message signing and individual signature verification is illustrated in Fig 2, based on the summary results from Table 4 and Fig 2, ECLAS has allover computation efficiency to the rest of the scheme except [13], and although it has slightly lower signing computation overhead it was found with security flaws in [23] whereas the proposed scheme satisfies the security requirements and withstand KGC escrow property " – page 15, lines 570-575. The text of the article needs to be revised.
  • Table 4 – the units of measurement used are not declared.
  • The same observation, the lack of units of measurement, for many of the relationships on page 15.

I am convinced that these corrections will not create any difficulties for the authors and that the paper will gain in clarity.

Author Response

Reviewer 1 Response

We appreciate for your precious time in reviewing our paper and providing valuable comments in a very time sensitive manner. We found the comments very valuable and insightful and have improved the manuscript to high standard.  After carefully taking account of the comments in responding we have a current version of the manuscript with the specific areas of corrections highlighted or changes described in the responses below. We hope that revision done meet the standard high for further processing of the manuscript. The authors welcome further constructive comments if any. Below we provide the point-by-point responses. All modifications in the manuscript have been highlighted in red.

Thank you for your consideration in all matters concerning this manuscript.

Open Review

English language and style

( ) Extensive editing of English language and style required
(x) Moderate English changes required
( ) English language and style are fine/minor spell check required
( ) I don't feel qualified to judge about the English language and style

Yes

Can be improved

Must be improved

Not applicable

Does the introduction provide sufficient background and include all relevant references?

( )

(x)

( )

( )

Is the research design appropriate?

( )

(x)

( )

( )

Are the methods adequately described?

( )

(x)

( )

( )

Are the results clearly presented?

( )

(x)

( )

( )

Are the conclusions supported by the results?

( )

(x)

( )

( )

Comments and Suggestions for Authors

The paper “Efficient Certificateless Aggregate Signature Scheme with Conditional Privacy-Preservation for Vecular Ad Hoc Networks Enhanced Smart Grid System” is presenting an interesting and complex problem for the future development of wireless communication technology of vehicles. However, I must make the following observations / comments:

  • In the title of the article a correction must be made, "Vehicular Ad Hoc Networks" instead of "Vecular Ad Hoc Networks".
  • The word Vehicular Ad Hoc Network is right corrected in the title.
  • The Figure 1 is not commented and discussed in the article’s text.
  • Reference to Figure 1., was made in “System Model”, in the line 204-211 of the section with the following further description made.
    • “The design and the interplay of these main entities in the system is illustrated in Fig. \ref{Figure1}, where close range network are facilitated by wireless communication technology such as IEEE802.11p. Mid-way network communication is aided by long range communication technology coupled with high bandwidth such as WiMax. Whereas, the backbone network system is empowered by wired communication which is mostly assumed to be secure as it controlled by the public utility company. It is the wireless communication that is supposed to be secured by security algorithm that ensures authentication and integrity on all communications amongst the concerned entities.”.

  • The phrase "Turing machine" (page 6, line 222) contains the name of a person, the English mathematician Alan Turing. Consequently, in the expression "Turing machine" the name Turing must be capitalized.
  • This observation was well noted and the correction right made as “Turing machine” and now it is in the line 232.

  • The Table 1 is not commented or discussed in the article’s text.
  • Reference to Table 1 was made under the preliminary section of the paper. Specifically, the statement added was “The notations used in the designed algorithm are given and described in Table \ref{table1}” in line 165.
  • Table 3 is included in the text before its description, which is not acceptable. The same observation for Table 4 and Table 5.     
    • This was well noted and corrections rightly applied, by ensuring table description is done first before it was referenced. This was done for Table 3 in line 551, Table 4 in line 583 and Table 5 in line 644.
    • For Table 3, Table 4 and Table 5, the issue was to do with latex typesetting and the figure location was effectively changed after this observation.

  • The authors use very long sentences, the meaning of which is difficult to understand. For example: "The computation cost for message signing and individual signature verification is illustrated in Fig 2, based on the summary results from Table 4 and Fig 2, ECLAS has allover computation efficiency to the rest of the scheme except [13], and although it has slightly lower signing computation overhead it was found with security flaws in [23] whereas the proposed scheme satisfies the security requirements and withstand KGC escrow property " – page 15, lines 570-575. The text of the article needs to be revised.
    • This was well noted and appreciated. To answer to this observation, we paraphrased of some parts of the sentences. This was corrected in lines 599-605. Specifically, the statements read as follows, “\textcolor{red}{Based on the generated summary results of computation cost comparison done in Table \ref{comparisontable} and the visual representation done given in Fig. \ref{Sign_Indi} we make conclusion on the performance of ECLAS. It is clear that the proposed ECLAS scheme, has allover computation efficiency to the rest of the scheme except \cite{cui2018efficient}, and although it has slightly lower signing computation overhead it was found with security flaws in \cite{kamil2019improved} whereas the proposed scheme satisfies the security requirements and withstand KGC escrow property.}”.
  • Table 4 – the units of measurement used are not declared.
  • The units for execution time for the cryptographic operations are micro-seconds (ms). This is well noted and corrections are made in Table 4 in line 605.
  • The same observation, the lack of units of measurement, for many of the relationships on page 15.
    • This was well noted as the units are inserted wherever it was required. The specific points edited were highlighted also in lines: 585, 587, 588, 595, 609, 610, 611 and 612.

I am convinced that these corrections will not create any difficulties for the authors and that the paper will gain in clarity.

Reviewer 2 Report

The authors design a secure and efficient certificate-less aggregate scheme (CLAS) for VANETs applicable in a smart grid scenario. The proposed scheme is based on elliptic curve cryptography to provide conditional privacy-preservation by incorporating usage of time validated pseudo-identification for communicating vehicles besides sorting out the key generation center escrow problem.

The paper is well written and it is scientifically sound. It brings contributions to the field of security in vehicular networks.

There are some issues that need to be addressed by the authors:

1- The term KCG is defined only in line 186. It is used several times before it's definition, including the abstract. You cannot assume all readers are familiar with the subject. I notice an abbreviation table, but each acronym needs to be defined the first time it is used in line.

2- There are several typos that need to be addressed such as the period between equations (2)-(3).

3- Section 4 needs to be properly indented. It is very hard to read it.

4- Captions for tables go over the table, not under.

Author Response

Reviewer 2 Response

We appreciate for your precious time in reviewing our paper and providing valuable comments in a very time sensitive manner. We found the comments very valuable and insightful and have improved the manuscript to high standard.  After carefully taking account of the comments in responding we have a current version of the manuscript with the specific areas of corrections highlighted or changes described in the responses below. We hope that revision done meet the standard high for further processing of the manuscript. The authors welcome further constructive comments if any. Below we provide the point-by-point responses. All modifications in the manuscript have been highlighted in red.

Thank you for your consideration in all matters concerning this manuscript.

Comments and Suggestions for Authors

The authors design a secure and efficient certificate-less aggregate scheme (ECLAS) for VANETs applicable in a smart grid scenario. The proposed scheme is based on elliptic curve cryptography to provide conditional privacy-preservation by incorporating usage of time validated pseudo-identification for communicating vehicles besides sorting out the key generation center escrow problem.

The paper is well written and it is scientifically sound. It brings contributions to the field of security in vehicular networks.

There are some issues that need to be addressed by the authors:

  • The term KGC is defined only in line 186. It is used several times before it's definition, including the abstract. You cannot assume all readers are familiar with the subject. I notice an abbreviation table, but each acronym needs to be defined the first time it is used in line.
    1. This observation was well noted and thereby the abbreviation KGC is now defined from where it is first used, that is in the abstract in line 10.

  • There are several typos that need to be addressed such as the period between equations (2)-(3).
    1. The typo was well corrected and the connecting phrase between equations (2) and (3) was revised to, “then, V_i computes,” in line 305.
  • Section 4 needs to be properly indented. It is very hard to read it.
    1. This was well noted and the section 4 representation was revised to ensure good readability by making it more spacious and more visible.
    2. The compacted equations were also aligned properly in the lines: 320 and 344
  • Captions for tables go over the table, not under.
    1. This was well noted and the captions for tables were revised to be on top not under the table in lines: 221, 538, 605, 613, 632 and 644.
